# Genetic screen identified PRMT5 as a neuroprotection target against cerebral ischemia

**Haoyang Wu[1,2†], Peiyuan Lv[1,2†], Jinyu Wang[1,2†], Brian Bennett[3], Jiajia Wang[4], Pishun Li[5], Yi Peng[1], Guang Hu[6]\*, Jiaji Lin[1,2]\***

[1]Department of Neurology, The Second Affiliated Hospital of Air Force Medical University, Xi'an, China; [2]Basic Medical School, Air Force Medical University, Xi'an, China; [3]Integrative Bioinformatics Support Group, National Institute of Environmental Health Sciences, Durham, United States; [4]Computer Network Information Center, Chinese Academy of Sciences, Beijing, China; [5]College of Veterinary Medicine, Hunan Agricultural University, Changsha, China; [6]Epigenetics and Stem Cell Biology Laboratory, National Institute of Environmental Health Sciences, Durham, United States

**\*For correspondence:**
hug4@niehs.nih.gov (GH);
jiaji90@outlook.com (JL)

†These authors contributed equally to this work

**Competing interest:** The authors declare that no competing interests exist.

**Abstract** Epigenetic regulators present novel opportunities for both ischemic stroke research and therapeutic interventions. While previous work has implicated that they may provide neuroprotection by potentially influencing coordinated sets of genes and pathways, most of them remain largely uncharacterized in ischemic conditions. In this study, we used the oxygen-glucose deprivation (OGD) model in the immortalized mouse hippocampal neuronal cell line HT-22 and carried out an RNAi screen on epigenetic regulators. PRMT5 was identified as a novel negative regulator of neuronal cell survival after OGD, which presented a phenotype of translocation from the cytosol to the nucleus upon oxygen and energy depletion both in vitro and in vivo. PRMT5 bound to the chromatin and a large number of promoter regions to repress downstream gene expression. Silencing *Prmt5* significantly dampened the OGD-induced changes for a large-scale of genes, and gene ontology analysis showed that PRMT5-target genes were highly enriched for Hedgehog signaling. Encouraged by the above observation, mice were treated with middle cerebral artery occlusion with the PRMT5 inhibitor EPZ015666 and found that PRMT5 inhibition sustains protection against neuronal death in vivo. Together, these findings revealed a novel epigenetic mechanism of PRMT5 in cerebral ischemia and uncovered a potential target for neuroprotection.

## eLife assessment

The authors performed a **useful** RNAi screen to identify epigenetic regulators involved in oxygen-glucose deprivation (OGD)-induced neuronal injury. PRMT5 was identified as a negative regulator of neuronal cell survival after OGD. **Solid** in vitro and in vivo data suggest that PRMT5 could be a novel therapeutic target for the treatment of ischemic stroke.

## Introduction

Ischemic stroke, which accounts for 87% of all strokes, is the second most common cause of death and the leading cause of acquired disability in adults worldwide (***Brott and Bogousslavsky, 2000***). However, current therapy represented by rapid circulation restoration only benefits ~5% of patients due to the narrow treatment window and small vessel blockage (***Brott and Bogousslavsky, 2000***).

There is an urgent need in developing neuroprotective drugs that provide an alternative strategy to minimize neural cell loss and maximize the potential for recovery, which finally extend the therapeutic window for recanalization and improve clinical outcomes after stroke. To develop such effective therapies, genomic approaches have been employed to elucidate the mechanism of ischemic stroke-induced brain damage and the underlying molecular basis. In particular, genome-wide association, genome sequencing, transcriptomic, proteomic, and metabolomic studies have identified many genomic loci, genes, and pathways that are associated stroke, providing potential diagnostic biomarkers and therapeutic targets (*Söderholm et al., 2019*; *Dichgans et al., 2019*). Intriguingly, increasing evidence suggests that epigenetic alterations play important roles in the pathogenesis of stroke (*Stanzione et al., 2020*). Epigenetic alterations are reversible changes in DNA or histone modifications, chromatin structure, and non-coding RNAs, and serve critical roles in the regulation of gene activity. For example, global DNA methylation, histone methylation, histone acetylation, microRNAs, enhancer RNAs, and circular RNAs were found to show significant changes after cerebral ischemia in animal models or human patients (*Stanzione et al., 2020*). Furthermore, inhibition of DNA methyltransferases or histone deacetylases by genetic or chemical mechanisms was shown to protect against stroke-induced brain damage (*Zhao et al., 2016*). Polycomb repressive complex genes were found to play important roles in neuroprotection after stroke (*Kuehner and Yao, 2019*). Thus, epigenetic regulators present novel opportunities for both stroke research and therapeutic interventions, and they may provide neuroprotection by potentially influencing coordinated sets of genes and pathways.

Both animal and cell culture models have been developed to facilitate the study of cerebral ischemia. In vivo, middle cerebral artery occlusion (MCAO) in rodents can produce reliable and well-reproducible infarcts and became a commonly used approach to mimic human ischemic stroke. In vitro, many cell-based models, using cells treated with chemical inhibitors, enzymatic induction, or hypoxia and energy depletion, were developed to replicate key features of ischemia (*Kurian and Pemaih, 2014*). Even though they cannot recapitulate the complex response of stroke in intact animals, they serve as convenient and amenable tools to study ischemia-reperfusion injuries via biochemical, genetic, and genomic methods. For example, oxygen-glucose deprivation (OGD) treatment has been widely used to examine the cellular mechanisms involved in cerebral ischemia and identify potential neuroprotective agents and pathways (*Bernstock et al., 2016*). In this study, we used the OGD model in the immortalized mouse hippocampal neuronal cell line HT-22 and carried out an RNAi screen on epigenetic regulators. We identified PRMT5 as a novel negative regulator of neuronal cell survival after OGD. We further showed that it promoted cell death by repressing genes involved in Hedgehog signaling, and its inhibition provides protection against ischemic injuries both in vitro and in vivo. Together, our findings revealed a novel epigenetic mechanism that contributes to neuronal cell death in cerebral ischemia and uncovered a potential target for neuroprotection.

## Results

### RNAi screen identified PRMT5 as a negative regulator of neuron survival after OGD

To systematically identify epigenetic factors that regulate cellular responses after cerebral ischemia, we used the well-established OGD protocol as an in vitro model for ischemic injury in the mouse hippocampal neuronal cell line HT-22. We carried out an RNAi screen using a custom shRNA library targeting selected epigenetic regulators. We first optimized the OGD treatment procedure on HT-22 cells and examined the length of OGD on cell survival. We chose 8 hr of oxygen and glucose deprivation followed by 24 hr of normal culture for the screen because it caused significant amount of cell death with small but consistent percentage of surviving cells. Next, we selected 125 genes involved in epigenetic and chromatin regulation and generated an shRNA library targeting these genes in the pLKO.1 vector (*Supplementary file 1*). To perform the screen (*Figure 1A*), we transduced HT-22 cells with the shRNA library, selected for puromycin-resistant cells, and split the cells into two equal populations. One population was frozen as a pellet and used later as the library-transduced cells. The other was treated with the optimized OGD procedure. After the treatment, the cells were cultured under normal conditions and expanded for another 10 d to allow those that survived OGD to grow. The OGD-treated cells were collected and frozen as a pellet as well. We extracted genomic DNA from both the library-transduced and the OGD-treated cell pellets, polymerase chain reaction

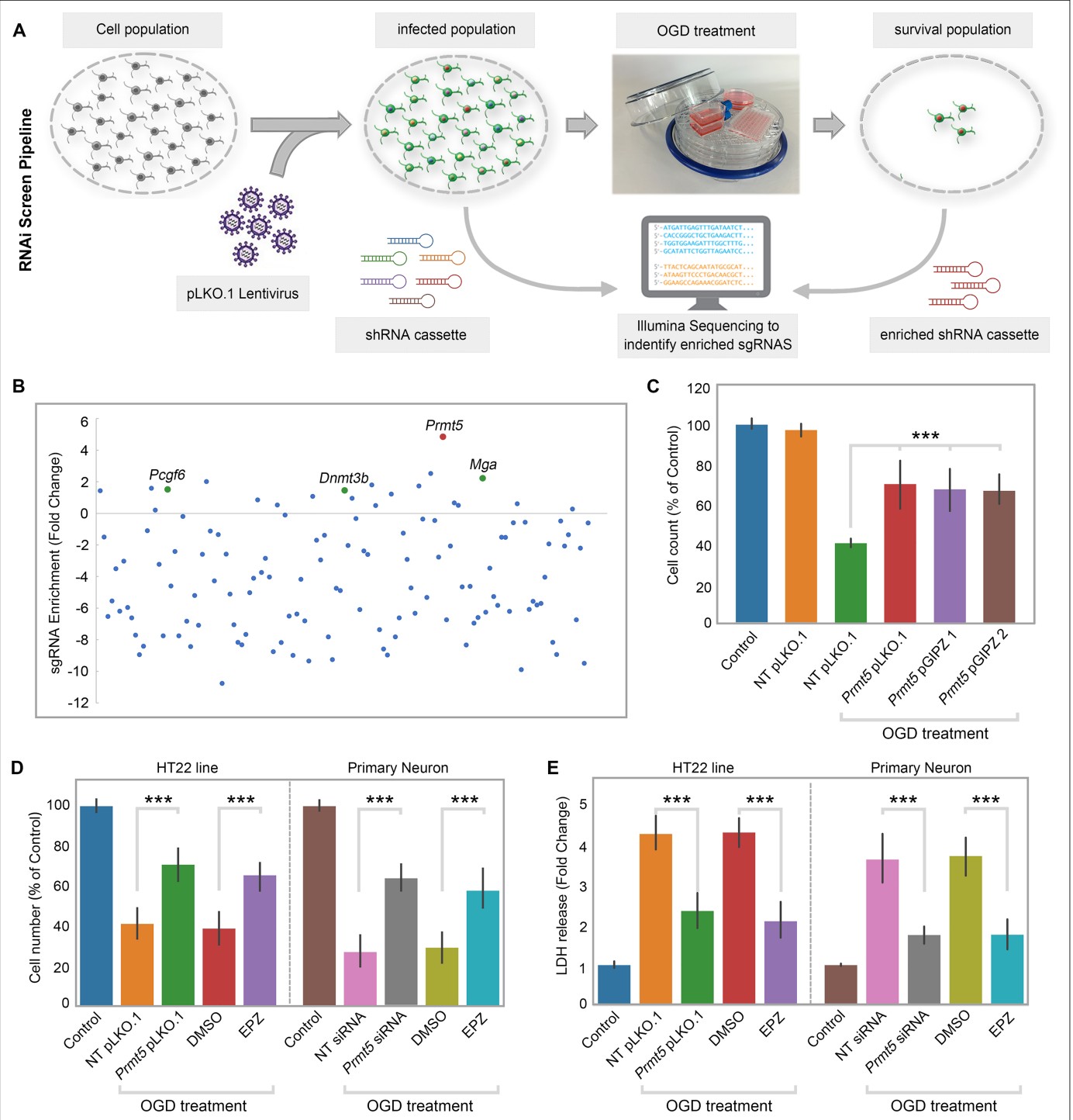

**Figure 1.** RNAi screen identified PRMT5 as a negative regulator of neuron survival after oxygen-glucose deprivation (OGD). (**A**) Overview of the study methodology. (**B**) shRNA representation after OGD treatment. *Prmt5* sgRNA have the highest enrichment after OGD treatment. (**C**) Effect of *Prmt5* silencing in HT-22 cells after OGD. HT-22 cells (Control) or HT-22 cells transduced with the indicated shRNA lentiviruses were cultured with or without OGD treatment. Cell numbers were counted after the treatment and normalized against the control. Compared with non-targeting shRNA lentiviruses (NT pLKO.1), *Prmt5* KD promoted a significant cell survival after OGD. (**D–, E**) Effect of *Prmt5* silencing or PRMT5 inhibition in HT-22 cells and primary neurons after OGD. Cell viability was determined using the Cell Counting Kit-8. Cellular cytotoxicity was determined using the Lactate Dehydrogenase Activity Assay Kit. Compared with NT pLKO.1, *Prmt5* pLKO.1 promoted a significant cell viability. Similar result could be found in PRMT5 inhibitor EPZ015666 (EPZ) vs. solvent control (DMSO). Graphing the mean with an SEM error bars, with ***p-value<0.001 by Student's *t*-test (n = 5).

(PCR)-amplified the shRNAs from the genomic DNAs, and sequenced the shRNA pools from the two cell populations using the Illumina platform. Based on the representation of the shRNAs in the cells, we found that most of the shRNAs were depleted after OGD treatment, suggesting that the genes they target are likely required for cell survival during or after OGD. Importantly, we also found a small number of shRNAs that showed increased representations after OGD, suggesting that the inhibition of their target genes likely promoted cell survival. Encouragingly, shRNAs against *Dnmt3b*, *Pcgf6*, and *Mga* were among those that are enriched after OGD treatment (*Figure 1B*), consistent with the notion that DNA methylation and polycomb repressive complexes play important roles in stroke injury and neuroprotection (*Endres et al., 2000*; *Dock et al., 2015*; *Elder et al., 2013*; *Chen et al., 2019*).

Among the enriched shRNAs, the shRNA against *Prmt5* displayed the largest fold change (*Figure 1B*). Therefore, we decided to focus on *Prmt5* and set out to validate its role in ischemic cell death. To rule out potential off-target effects by the *Prmt5* shRNA in the screen, we designed additional shRNAs against *Prmt5* in a different vector pGIPZ. We found that *Prmt5* knockdown (KD) by the original pLKO.1 shRNA as well as the two new pGIPZ shRNAs all led to significant improvement in HT-22 cell survival after OGD (*Figure 1C*). Moreover, treatment of the cells with a small-molecule PRMT5 inhibitor EPZ015666 resulted in similar outcomes (*Figure 1D*), further supporting the fact that PRMT5 is responsible for the protective effect. Consistent with the above results based on cell numbers, PRMT5 inhibition significantly reduced OGD-induced HT-22 cell death in the lactate dehydrogenase (LDH) cytotoxicity assay (*Figure 1E*). Finally, in addition to the immortalized HT-22 cell line, *Prmt5* shRNA or PRMT5 inhibitor promoted the survival of primary mouse neurons in culture after OGD treatment (*Figure 1D–E*). Together, these results clearly demonstrated that PRMT5 plays an important role during OGD and its inhibition protects neuronal cells against OGD-induced damage.

## PRMT5 protein translocates into the nucleus upon ischemic injury

To understand the molecular function of PRMT5 in OGD, we examined the behavior of its protein product by imaging in HT-22 cells. It was found that PRMT5 is predominantly localized in the cytoplasm under normal conditions. Upon OGD treatment, however, it relocates to the nucleus and becomes enriched in the nuclear compartment (*Figure 2A*). To rule out any potential artifacts caused by the antibody, we knocked-in an HA-tag to the C-terminus of the endogenous *Prmt5* gene in HT-22 cells using CRISPR-mediated genome editing, and re-examined PRMT5 localization with the HA antibody. Similarly, we found that the endogenous HA-tagged PRMT5 translocates from the cytosol to the nucleus after OGD (*Figure 2A*). To complement the imaging results, we carried out biochemical fractionation of the HT-22 cells and separated the cellular content into cytoplasmic, nucleoplasmic, and chromatin-bound fractions. Consistent with the above, PRMT5 protein moved from the cytoplasm to the nucleus in OGD-treated cells (*Figure 2B*). Further, we found that PRMT5 protein became largely chromatin-bound, suggesting that it may participate in OGD-induced transcriptional responses. Finally, we repeated the experiments in OGD-treated primary mouse neurons in culture and observed the same result (*Figure 2C and D*).

Next, we wanted to test whether PRMT5 translocation also happens in vivo. The left MCAO model was applied to mimic ischemic injury in the mouse brain and examined PRMT5 localization in both the ipsilateral and contralateral side by immunohistochemistry. Similar to what was observed in cultured cells, PRMT5 signal was mostly detected in the cytoplasm on the contralateral side, but became highly enriched in the nucleus on the ipsilateral side (*Figure 3A*). Upon closer examination, PRMT5 nuclear translocation was detected in both the penumbra area and ischemic core of the ipsilateral side (*Figure 3B*). We further verified these observations by immunofluorescence staining (*Figure 3C*). Importantly, biochemical fractionation of the neuronal tissues again showed that PRMT5 not only translocates into the nucleus but also binds to the chromatin upon ischemic injury (*Figure 3D*). As PRMT5 has been implicated in chromatin and gene regulation, these results strongly suggested that PRMT5 may enter the nucleus and bind the chromatin to regulate neuronal gene expression upon OGD.

## PRMT5 represses Hedgehog signaling expression after OGD

To test whether and how PRMT5 regulates gene expression after OGD, we carried out RNA-seq in OGD-treated HT-22 cells vs. control, as well as OGD-treated HT-22 cells with *Prmt5* KD vs. control. We found that OGD treatment led to the upregulation of 2186 genes and downregulation of 2926

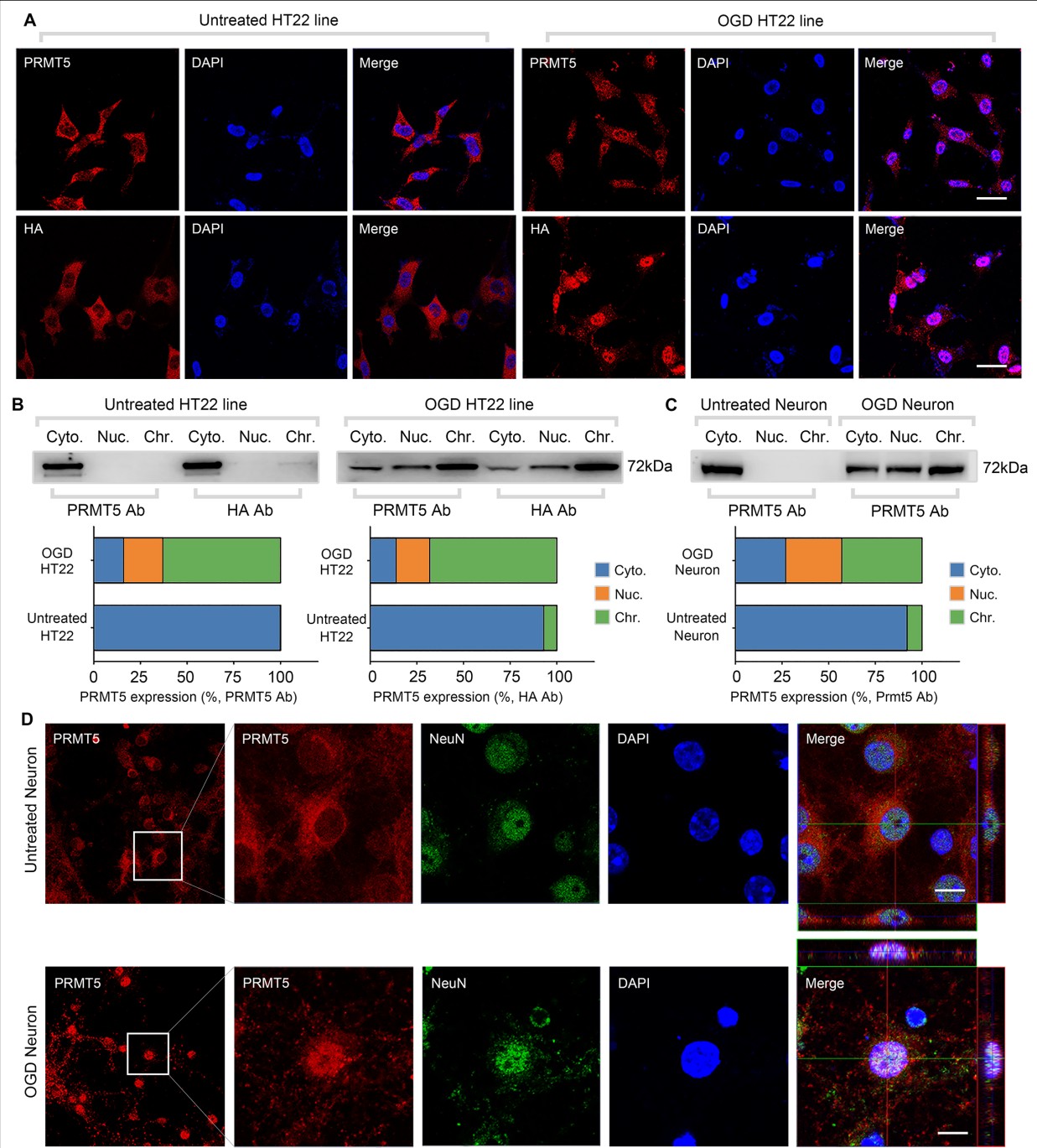

**Figure 2.** Oxygen-glucose deprivation led to PRMT5 nuclear translocation in vitro. (**A**) Immunofluorescence staining to show PRMT5 cellular localization after oxygen-glucose deprivation (OGD) in HT-22 cells. HT-22 cells were stained with the PRMT5 antibody or HA antibody for C-terminal HA-tagging of the endogenous PRMT5 by CRIPR-mediated genome editing. Cell nuclei were counterstained by DAPI. Scale bar = 20 µm. (**B**) Biochemical fractionation to show PRMT5 cellular localization after OGD. PRMT5 in each fraction was detected by western blot using either the PRMT5 or HA antibody. (**C, D**) Primary cortical neurons were treated with or without OGD and subjected to biochemical fractionation. Primary neurons were stained with the PRMT5 or NeuN (neuronal marker) antibody. Scale bar = 8 µm. Cyto., Nuc., and Chr. stand for cytoplasmic, nucleoplasmic, and chromatin-bound fractions.

The online version of this article includes the following source data for figure 2:

**Source data 1.** Source data for the western blot analysis in the left panel of *Figure 2B* (anti-PRMT5 and anti-HA).

**Source data 2.** Source data for the western blot analysis in the right panel of *Figure 2B* (anti-PRMT5 and anti-HA).

**Source data 3.** Source data for the western blot analysis in *Figure 2C* (anti-PRMT5).

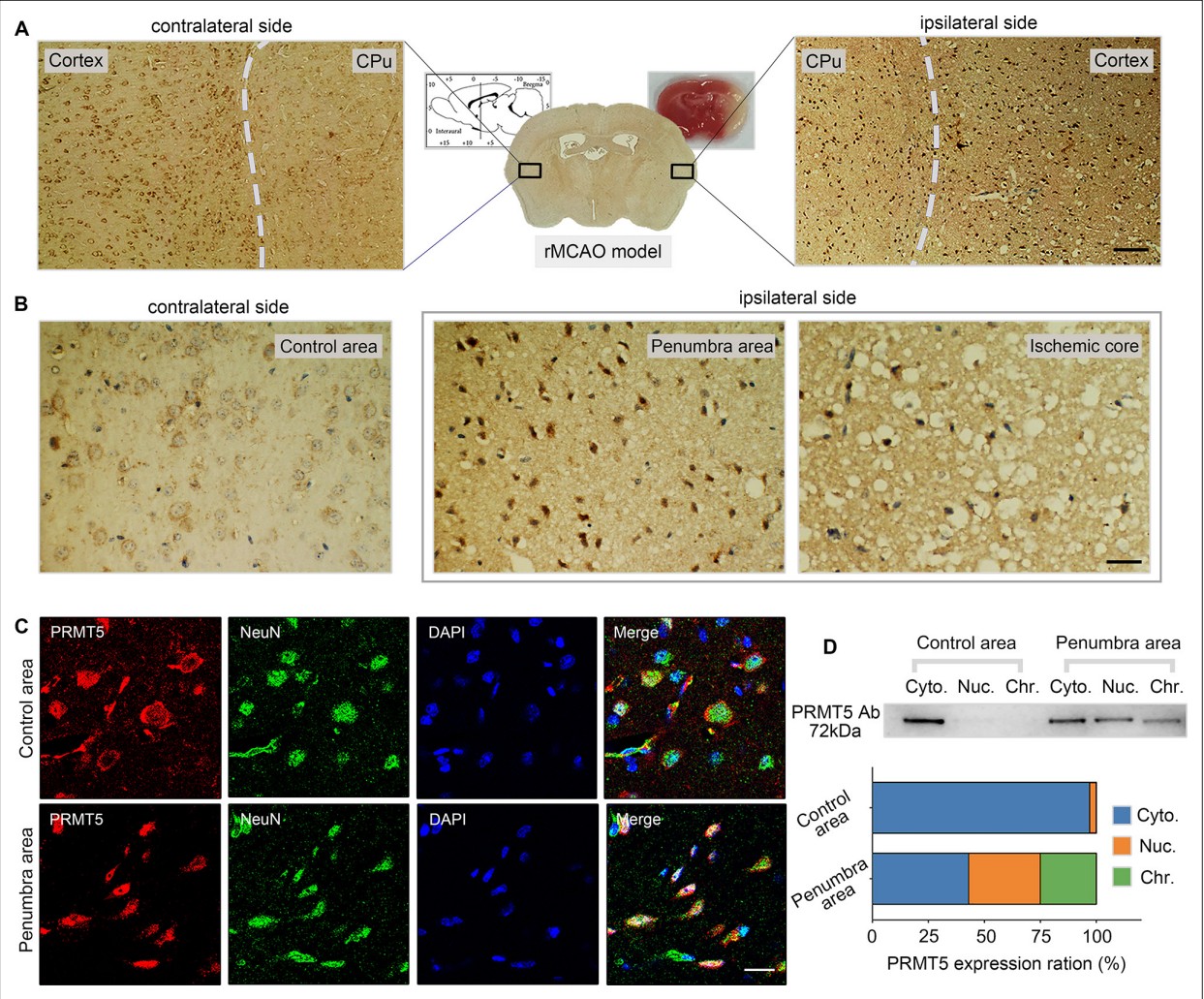

**Figure 3.** Middle cerebral artery occlusion led to PRMT5 nuclear translocation in vivo. (**A–C**) Immunohistochemical and immunofluorescence staining to show PRMT5 cellular localization in ipsilateral and contralateral sides after middle cerebral artery occlusion (MCAO). Scale bar = 3 μm, 9 μm and 20 μm for A/B/C. (**D**) Biochemical fractionation to show PRMT5 cellular localization after MCAO. PRMT5 in each fraction was detected by western blot using the PRMT5 antibody. Cyto., Nuc., and Chr. stand for cytoplasmic, nucleoplasmic, and chromatin-bound fractions.

The online version of this article includes the following source data for figure 3:

**Source data 1.** Source data for the western blot analysis in *Figure 3D* (anti-PRMT5).

genes (*Figure 4A*, *Supplementary file 2*). Interestingly, while *Prmt5* KD alone had minimal impact on gene expression, it significantly dampened the OGD-induced changes for a large number of genes (*Figure 4B*, *Supplementary file 2*). Specifically, 1035 of the 2186 OGD-induced genes and 2090 of the 2926 OGD-repressed genes were reversed or partially reversed upon *Prmt5* KD (*Figure 4B*, *Supplementary file 2*). This is consistent with the cellular phenotype, in which *Prmt5* KD partially rescued OGD-induced cell death.

As shown in *Figure 2*, OGD treatment resulted in PRMT5 nuclear translocation and chromatin binding. To test whether PRMT5 directly regulates gene expression and contributes to OGD-induced transcriptional changes, we carried out PRMT5 chromatin immunoprecipitation followed by high-throughput sequencing (ChIP-seq) in HT-22 cells treated with OGD. In total, we identified 16,061 peaks bound by PRMT5. Interestingly, a significant fraction of the PRMT5-bound genomic regions (56%) are near the transcription start sites (TSSs) (*Figure 4C*, *Supplementary file 2*), consistent with the notion that PRMT5 may regulate downstream gene expression. Indeed, when intercepted with the RNA-seq data, we found that PRMT5 occupies the promoter regions of a large fraction of the 2090 genes that were down-reversed genes in OGD (*Figure 4D*, *Supplementary file 2*). Furthermore,

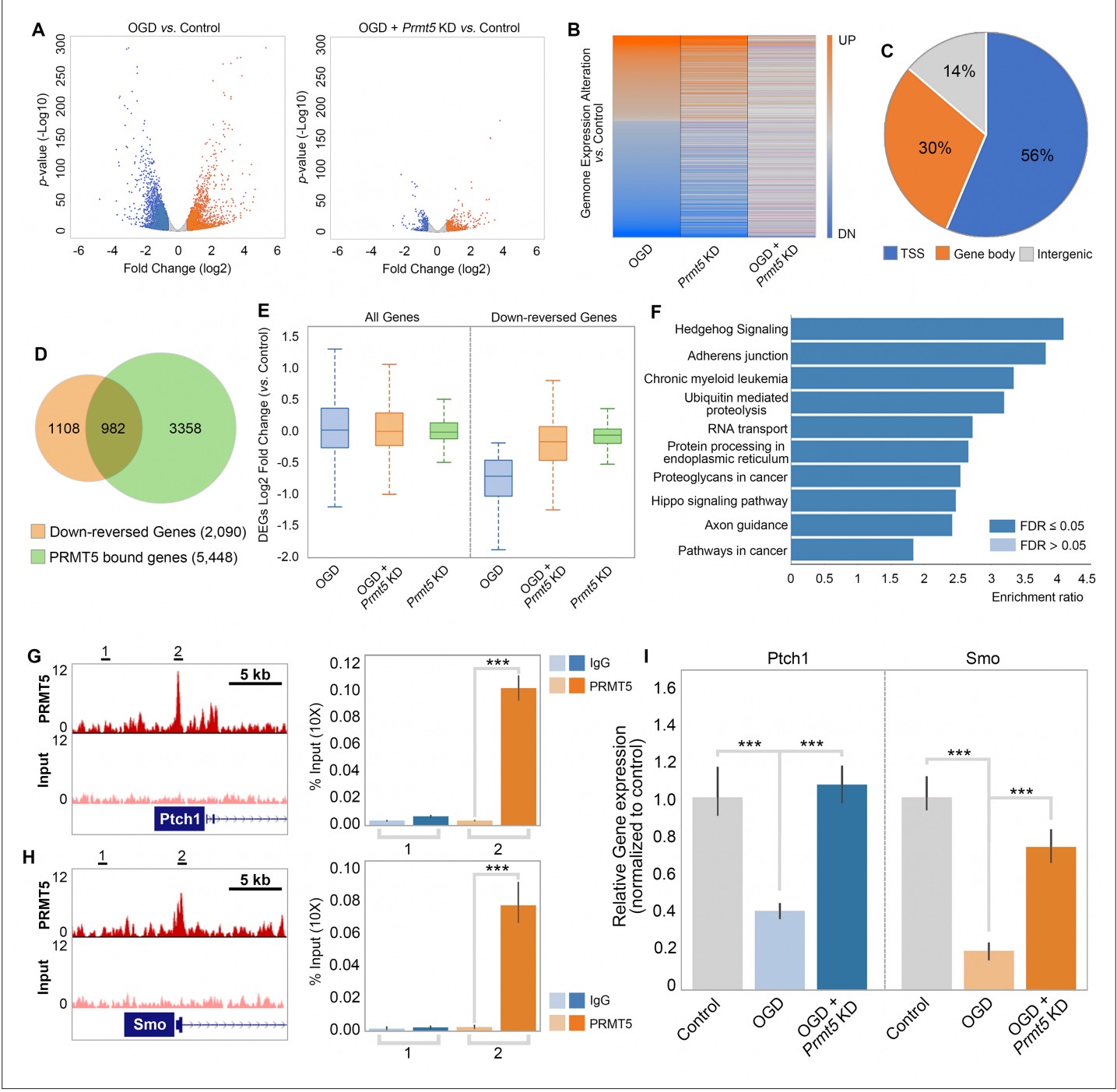

**Figure 4.** PRMT5 represses Hedgehog signaling expression after oxygen-glucose deprivation (OGD). (**A**) Gene expression changes after OGD. HT-22 cells were transduced with Control (non-targeting) or *Prmt5* shRNA lentiviruses, drug selected, and treated with OGD. Cells were collected and total RNAs were extracted for RNA-seq. Differentially expressed genes (DEGs) after OGD in the control cells were identified by log2-fold change >1 and p-value<0.01. Orange: upregulated genes after OGD; blue: downregulated genes after OGD. (**B**) Effect of *Prmt5*-KD on OGD-induced gene expression changes. OGD-induced DEGs were defined as described in (**A**), and the expression changes of the DEGs after OGD in control and *Prmt5*-KD cells were plotted by heat map. (**C, D**) Genomic regions occupied by PRMT5 based on ChIP-seq in HT-22 cells treated with OGD. Reversal of OGD-induced gene expression changes by *Prmt5*-KD. PRMT5-target genes were defined as described in the text. (**E**) Expression changes of all detected genes or PRMT5-target genes after OGD in control or *Prmt5*-KD cells were examined by box plots. (**F**) Kyoto Encyclopedia of Genes and Genomes (KEGG) pathway analysis of PRMT5-target genes. (**G, H**) Genome browser tracks of PRMT5 occupancy at selected PRMT5-target genes. ChIP-qPCR of PRMT5 occupancy in HT-22 cells treated with OGD at selected PRMT5-target genes (*Ptch1* and *Smo*). (**I**) RT-qPCR to show the expression of selected PRMT5-target genes in control or *Prmt5*-KD HT-22 cells upon OGD treatment. Graphing the mean with an SEM error bars, with ***p-value<0.001 by Student's *t*-test (n = 5).

the OGD-repressed genes appeared to be dependent on PRMT5 as *Prmt5* KD largely reversed their expression changes in OGD (*Figure 4E*). Therefore, we defined these genes that are bound by PRMT5, downregulated after OGD treatment but rescued by *Prmt5* KD as PRMT5-target genes in OGD. And the above results suggested that under OGD conditions PRMT5 enters the nucleus and suppresses the expression of its target genes. Gene ontology analysis showed that PRMT5-target genes are highly enriched for Hedgehog signaling (*Figure 4F*). To validate the above findings based on genomic data, we carried out ChIP quantitative PCRs (ChIP-qPCRs) as well as reverse transcription quantitative PCRs (RT-qPCRs) on selected OGD-repressed PRMT5-target genes, including *Ptch1* and *Smo*. It was found that PRMT5 indeed occupied their promoter regions (*Figure 4G and H*) and is responsible for their downregulation after OGD treatment (*Figure 4I*). Together, our findings strongly supported the notion that upon oxygen and glucose deprivation, PRMT5 promotes neuronal cell death by repressing the expression of genes involved in Hedgehog signaling. *Prmt5* silencing provides neuroprotection by preventing the inhibition of Hedgehog pathways.

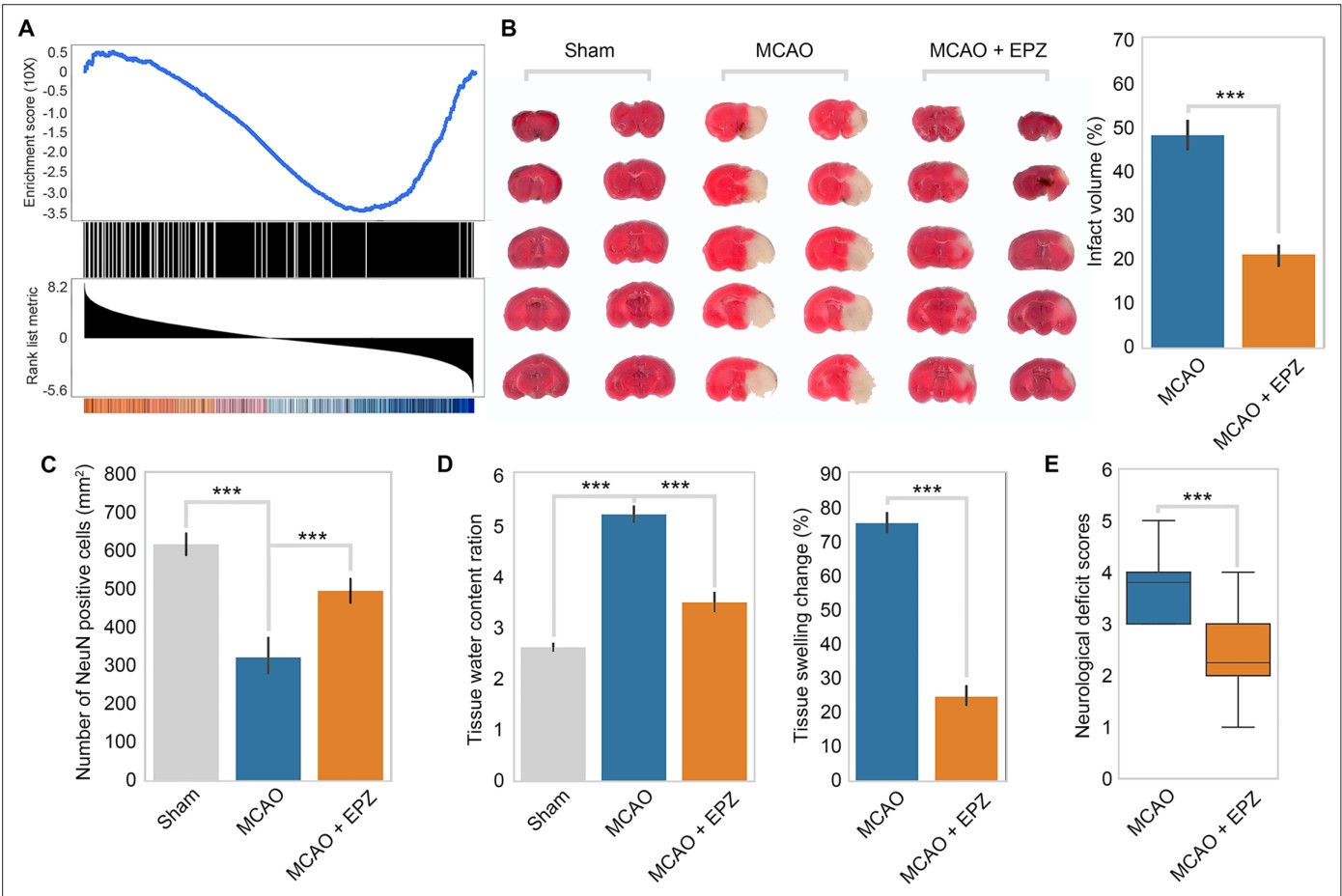

**Figure 5.** PRMT5 inhibition promotes neuron survival after ischemic injury in mouse brain. (**A**) Gene set enrichment analysis for PRMT5-target genes in the middle cerebral artery occlusion (MCAO) mouse stroke model. Gene expression changes after MCAO was based on 29862458. (**B**) Effect of PRMT5 inhibition on infarct volume in the MCAO model. Mice were treated with subjected with sham (Sham group), MCAO with solvent (MCAO group), or MCAO with the PRMT5 inhibitor EPZ015666 (MCAO + EPZ group). The infarct volume was determined by 2,3,5-triphenyltetrazolium chloride (TTC) staining. (**C–E**) For tissue immunofluorescence staining, samples were fixed using 4% paraformaldehyde at room temperature for 15 min, followed by 0.5% Triton X-100 permeabilization for 10 min and 0.5% bovine serum albumin blocking for 30 min. The mice were sacrificed for NeuN immunofluorescence staining, and positive staining cell were count as survival neurons. Tissue water content ration, tissue swelling change, and neurological deficit score were determined as described in 'Materials and methods'. Graphing the mean with an SEM error bars, with ***p-value<0.001 by Student's *t*-test (n = 12).

# PRMT5 inhibition promotes neuron survival after ischemic injury in mouse brain

To test the relevance of our model in vivo, we first examined the behavior of the PRMT5-target genes in OGD in ischemic injury in mice. We used a public RNA-seq dataset (GSE112348) generated from an MCAO mouse model and carried out gene set enrichment analysis (GSEA). We found that PRMT5-target genes in OGD are highly enriched for those that were downregulated after cerebral infarction in the MCAO mouse (*Figure 5A*), suggesting that our model may apply in animals.

Encouraged by the above observation, we set out to test whether PRMT5 inhibition may provide protection against neuronal cell death in mouse brain after ischemic injury as it did in cultured cells. We treated adult mice with either solvent only or the PRMT5 inhibitor EPZ015666 intranasally and then induced ischemic injury in the left side of the brain with the MCAO procedure. After the surgery, we first assessed the neurological damage based on behavioral changes and then sacrificed the animals and examined the damages in their brain tissue. Indeed, we found that PRMT5 inhibition resulted in significant protective effects. Specifically, EPZ015666 treatment reduced the infarct volume (*Figure 5B*) and increased the number of NeuN-positive neurons after MCAO (*Figure 5C*). It also reduced the water content and tissue swelling caused by infarction (*Figure 5D*). Most importantly, we found that EPZ015666 treatment led to significantly improved functional recovery in the animals as evidenced by the reduction in the neurological deficit scores (*Figure 5E*). Therefore, these results strongly suggested that PRMT5 plays a critical role in mediating neuronal cell death after ischemic injury in the brain and may serve as a novel therapeutic target for stroke treatment.

## Discussion

While genome-wide association and expression profiling studies have uncovered many genomic loci, genes, and regulatory elements involved in stroke, the causal factors underlying stroke-induced damage are often confounded by the large number of incidental events accompanying stroke. Therefore, new approaches to define genes with functional relevance are essential to understand stroke and identify potential therapeutic targets. In that sense, genetic screens can provide a more direct strategy to search for genes that support neuronal cell survival during stroke conditions (*Bernstock et al., 2016*). The development of RNAi and CRIPSR technologies, as well as the multiplex screening method, further promoted fast and convenient screens in mammalian cells (*Sioud, 2020*). Here, we carried out a small-scale RNAi screen in HT-22 cells and identified a list of epigenetic factors that regulate neuronal cell death under ischemic conditions. Our study illustrates the power of forward genetics for the systematic study of ischemic stroke-induced injuries and presents a new strategy for the identification of neuroprotection agents.

Among the hits from the screen, we focused on *Prmt5* as its silencing presented the strongest phenotype. PRMT5 belongs to the evolutionarily conserved arginine methyltransferase protein family (*Dong et al., 2022*). It catalyzes the transfer of a methyl group from S-adenosyl-methionine to the guanidino nitrogen atoms of arginine as monomethylation and symmetric demethylation (*Dong et al., 2022*). PRMT5 has been implicated in various developmental processes and diseases (*Motolani et al., 2021*). In particular, it is overexpressed in many cancers and elevated PRMT5 expression often correlates with poor prognosis (*Kim and Ronai, 2020*). Conversely, its depletion or inhibition leads to reduced proliferation and increased apoptosis in cancer cells (*Kim and Ronai, 2020*). In addition, PRMT5 is also required for early embryonic development and the maintenance of embryonic and neural stem cells (*Chittka et al., 2012*). However, we found that *Prmt5* silencing by RNAi or inhibition by a small molecule enhances neuronal cell survival upon ischemic injury both in vitro and in vivo. Consistent with our findings, *Prmt5* silencing was reported to reduce oxidative stress-induced cell death in renal ischemia (*Diao et al., 2019*). Thus, PRMT5 appears to play dual roles in cell viability in a context-dependent fashion, being pro-proliferation in cancer and stem cells but pro-cell death under ischemic conditions.

Mechanistically, we showed that PRMT5 protein translocates from the cytosol to the nucleus upon oxygen and energy depletion both in vitro and in vivo. Like other arginine methyltransferase family members, subcellular localization of PRMT5 protein appears to be cell type specific (*Koh et al., 2015*). In embryonic stem cells, it is predominantly cytoplasmic in self-renewing condition, but relocates to the nucleus upon differentiation (*Gkountela et al., 2014*). During mammalian embryonic

development, it shuttles between the cytosol and the nucleus to fulfill its many functions in gene regulation and epigenome silencing (*Ancelin et al., 2006*; *Kim et al., 2014*). In cancer, PRMT5 protein was found to show different cellular localization between normal and tumor tissues and between tumor types (*Gu et al., 2012*; *Nicholas et al., 2013*), suggesting that its compartment-specific functions may regulate distinct molecular programs. In our study, we showed that PRMT5 is highly expressed in neurons and localizes predominantly in the cytoplasm under normal conditions, consistent with previous reports (*Chittka, 2013*; *Ratovitski et al., 2015*). Under OGD or ischemic conditions, it relocates to the nucleus and contributes to neuronal cell death. As transcription factors can interact with PRMT5 and regulate its cellular localization (*Shilo et al., 2013*), we speculate that PRMT5 may be recruited to the nucleus to carry out its function in transcriptional regulation by factor(s) in response to ischemia.

PRMT5 plays important roles in gene regulation by methylating histones, transcription factors, and spliceosome proteins (*Blanc and Richard, 2017*). It also interacts with ATP-dependent chromatin remodelers to control gene expression (*Dacwag et al., 2007*). We found that as PRMT5 translocates to the nucleus upon OGD, it binds to the chromatin and a large number of promoter regions to repress downstream gene expression. This is consistent with the notion that PRMT5 often acts as a transcription repressor (*Fabbrizio et al., 2002*). Interestingly, OGD treatment alone caused similar numbers of genes to be up- or downregulated. However, PRMT5-target genes are mostly downregulated after OGD and are enriched for those involved in cell survival. Thus, OGD or stroke-induced neuronal cell death may largely be caused by the repression of survival pathways and re-activating these pathways may thereby provide an important mechanism for neuroprotection.

Hedgehog signaling is an important signaling pathways that is frequently used during development for intercellular communication, as well as regeneration and homeostasis (*Carballo et al., 2018*). In nervous system-specific hedgehog, sonic hedgehog (SHH), canonically binds and inactivates PTCH1 inhibition of SMO for downstream GLI-1 signaling cascade, which has particularly marked roles in nervous system cell-type specification and limbs patterning (*Briscoe and Ericson, 2001*; *Dessaud et al., 2008*). Many studies have emphasized its impact on neural cell survival and tissue regeneration/repair after ischemic stroke. *Chechneva et al., 2014* found that SHH signaling activation has promoted on neural cell survival and tissue regeneration/repair after ischemic stroke. Moreover, SHH agonist could also induce a decrease in blood–brain barrier permeability and an increase of newly generated neurons and neovascularization in the MCAO model (*Chechneva et al., 2014*). In in vitro studies, the expression of SHH, PTCH1, and GLI-1 was significantly downregulated within 24 hr after OGD exposure, which were closely associated with increasing numbers of apoptotic cells (*Yin et al., 2020*). Activation of SHH signals promoted CREB and AKT phosphorylation, upregulated the expressions of BDNF, neuroligin, and neurexin, and decreased NF-κB phosphorylation following OGD (*Yin et al., 2020*). Therefore, the regulation of PRMT5 on SHH signaling may be one of the important mechanisms of its ischemia protection.

As mentioned above, PRMT5 plays critical roles in tumorigenesis and is therefore a therapeutic target that is actively pursued for cancer treatment. Several PRMT5 inhibitors, some of which highly potent, are in clinical trials to test their safety and efficacy (*Tao et al., 2019*). It will be interesting to systematically test their effect in protecting against neuronal cell death in cell-based and animal models of stroke in order to gain more insights to the molecular events after stroke as well as develop potential therapeutic interventions.

There were two major limitations that should be contemplated when interpreting these results. On the one hand, HT-22 is an immortalized mouse hippocampal cell line, which is a subline derived from parent HT4 cells that were originally immortalized from cultures of primary mouse hippocampal neurons (*Pascual et al., 2012*). There exists to a certain extent difference from primary neurons under OGD stress and may be affected by cell proliferation regulatory elements in HT-22. On the other hand, the PRMT5 function seems to closely link to its subcellular localization. The nuclear PRMT5 plays specific roles in transcription regulation by directly modulating the activity of several transcription factors or by methylating histones (H4R3, H2AR3, H3R8, and H3R2), while the cytoplasmic expression of PRMT5 is involved in splicing, translation, and regulation of receptor–ligand signaling and organelle integrity (*Koh et al., 2015*). The mechanism of how PRMT5 is involved in neuroprotection needs further research.

## Materials and methods

### HT-22 cells and primary cortical neurons culture

HT-22 cells were kindly provided by Dr. Richard Dargusch (Salk). Its identity has been authenticated with short tandem repeat (STR) profiling, and no mycoplasma or other microbiological contamination was found. Cells were routinely cultured in Dulbecco's Modified Eagle's Medium (Gibco) supplemented with 10% fetal bovine serum (Gibco), and the cultures were maintained at 37°C in a humidified atmosphere containing 5% $CO_2$. For CRISPR-mediated gene targeting, pX330 and homologous recombination (HR) donor plasmids were co-transfected into HT-22 for HA-tag knock-in. Transfected cells were seeded at colonal density, and individual colonies were picked and screened by PCR. Correctly targeted clones were amplified and re-screened to confirm genotype. The primers and oligos used in this study are listed in *Supplementary file 3*. The neonatal mouse neurons were isolated as previously described with modifications (*Dessaud et al., 2008*). Briefly, cerebral cortices were removed from 1- to 3-day-old C57BL/6 mouse pups, stripped of meninges and blood vessels, and minced. Tissues were dissociated with 0.25% trypsin for 15 min at 37°C and gentle trituration. Neurons were resuspended in a complete culture medium (Neurobasal medium containing 2% B27 supplement and 0.5 mM L-glutamine) and plated at a density of $3 \times 10^5$ cells/cm$^2$. Neurons were maintained at 37°C in a humidified 5% $CO_2$ incubator and half of the culture medium was changed every other day. The cultured neurons were used for experiments within 8–10 d. All experimental protocols and animal handling procedures were performed in accordance with the National Institutes of Health (NIH) guidelines for the use of experimental animals and were approved by the Animal Care and Use Committee of The Second Affiliated Hospital of Air Force Medical University (IR-2384-3915-3322).

### OGD treatment

OGD treatment was carried out as previously described (*Goldberg and Choi, 1993*). Briefly, cells were rinsed and cultured in the glucose-free Earle's balanced salt solution (BSS) with the following composition (in mmol/L): 116 NaCl, 5.4 KCl, 0.8 $MgSO_4 \cdot 7H_2O$, 1 $NaH_2PO_4 \cdot 2H_2O$, 26.2 $NaHCO_3$, 0.01 glycine, 1.8 $CaCl_2 \cdot 2H_2O$, and pH 7.4. The culture was then placed in an Hypoxia Incubator Chamber (StemCell Technology) and were equilibrated for 15 min with a continuous flux of 95% $N_2$/5% $CO_2$ gas. The chamber was then sealed and placed into a humidified incubator at 37°C for 8 hr (HT-22) or 60 min (primary cortical neuron). OGD was terminated by removing the cultures from the chamber, replacing BSS with normal culture medium, and returning cells to the normoxic conditions for 24 hr. Cells cultured under normal conditions during the experimental period were used as controls.

For PRMT5 inhibitor treatment, EPZ015666 (Selleckchem Company) was prepared in dimethyl sulfoxide (DMSO) at a concentration of 100 mM and stored at –20 °C before use. EPZ (10 µM final concentration) or DMSO was added to cells 20 min before the initiation of OGD and to the BSS during the OGD treatment. For siRNA transfection of the primary neurons, the culture medium was refreshed at day 7. Cells were transfected with siRNAs using Lipofectamine 3000 according to the manufacturer's instructions. The medium was replaced the next day, and OGD experiments were performed 48 hr after transfection.

### shRNA library and RNAi screen for OGD

The oligos used in this study are listed in *Supplementary file 3*. For pLKO.1 shRNAs, complementary single-stranded oligos were annealed and cloned as suggested by the RNAi consortium. pGIPZ shRNAs were ordered from Dharmacon. All plasmids were confirmed by sequencing. The shRNA library used for the screen was as previously described (*Li et al., 2017*). shRNA lentivirus was prepared by transfecting 293T cells using TransIT-293 (Mirus) using a standard protocol from Addgene.

For the shRNA screen, HT-22 cells were transduced with shRNA lentivirus and selected for viral infection with 2 µg/mL puromycin for 4 d. After selection, cells were split into two halves. One half was harvested and frozen as a pellet (untreated). The other half was subjected to OGD treatment, replated, cultured until confluent, and harvested (treated). Genomic DNA was extracted from the cell pellets, and the integrated shRNAs were amplified according to the pooled screen protocol from the Broad Institute (https://portals.broadinstitute.org/gpp/public/resources/protocols). The amplified shRNAs were sequenced on the Illumina platform, and the representations of the shRNAs in the untreated and treated cells were calculated to identify shRNAs that were enriched after treatment.

## RT-qPCR and RNA-seq

Total RNA was isolated from cells using the GeneJet RNA purification kit (Thermo Scientific), and 0.5 µg total RNA was reverse transcribed to generate cDNA using the iScript cDNA Synthesis Kit (Bio-Rad) according to the manufacturer's instructions. RT-qPCRs were performed using the SsoFast EvaGreen Supermix (Bio-Rad) on the Bio-Rad CFX-384 Real-Time PCR System. Actin was used for normalization. All experiments were performed at least three times, and representative results are shown in the figures. For RNA-seq, libraries were prepared from two biological replicates using the TruSeq RNA Sample Prep Kit and sequenced on the NextSeq (Illumina).

## Cell viability and LDH assay

Cell viability assay was performed using the cell proliferation reagent Cell Counting Kit-8 (CCK-8) following the manufacturer's protocol (Sigma). Briefly, cells were cultured in 96-well plates and subjected to different treatments as described in each experiment. Then, 10 µL CCK-8 was added to each well, and the plates were incubated at 37°C for 90 min. The absorbance was determined with a Microplate Reader (Bio-Rad). Neuronal cytotoxicity was determined with the Lactate Dehydrogenase Activity Assay Kit according to the manufacturer's instructions (Sigma). Briefly, cells were cultured in 96-well plates and subjected to different treatments as described in each experiment. Then, 50 µL of supernatant from each well was collected to assay the LDH activity.

## Subcellular fractionation and western blot

Subcellular fractionation was adapted from *Méndez and Stillman, 2000*. Briefly, harvested cells were washed with phosphate-buffered saline (PBS) and resuspended in cytosolic buffer (10 mM HEPES [pH 7.4], 10 mM KCl, 1.5 mM MgCl$_2$, 0.34 M sucrose, 10% glycerol, 1 mM dithiothreitol [DTT]) supplemented with protease inhibitors at a concentration of 20–40 million cells/mL, and incubated on ice for 5 min. A total of 1% Triton-X 100 in equal volume of cytosolic buffer was added to a final concentration of 0.1%, and the cells were mixed by gently pipetting and further incubated for 10 min on ice. Ten percent of the cell suspension was taken as the whole-cell extract fraction. The remaining cell suspension was centrifuged at 1300 × *g* for 5 min at 4°C to separate the cell nuclei, and supernatant containing the cytoplasmic fraction was collected. Nuclei were washed once in cytosolic buffer, then lysed 10 min on ice in 1× volume chromatin extraction buffer (3 mM EDTA, 0.2 mM EGTA, 1 mM DTT) supplemented with protease inhibitors. Insoluble chromatin was pelleted by centrifugation at 1700 × *g* for 5 min at 4°C, and supernatant containing the nucleoplasm fraction was collected. The chromatin pellet was washed once with chromatin extraction buffer. All fractions were boiled in LDS loading buffer. All experiments were performed three or more times, and representative results are shown in the figures.

Cell lysates or extractions were loaded into a NuPAGE Bis-Tris gels (4–12%) and transferred onto PVDF or NC membranes. The membrane was blocked with 5% non-fat milk, followed by incubation with primary antibodies (*Supplementary file 4*) and HRP-conjugated secondary antibodies. Chemiluminescence signal was generated with ECL reagents (GE) and detected using ChemiDoc Touch Imaging System (Bio-Rad).

## Immunostaining

For cell immunofluorescence staining, samples were fixed using 4% paraformaldehyde at room temperature for 15 min, followed by 0.5% Triton X-100 permeabilization for 10 min and 0.5% bovine serum albumin blocking for 30 min. Samples were then incubated with primary antibodies (*Supplementary file 4*) at 37°C for 2 hr or 4°C overnight, followed by fluorescent secondary antibodies (Life Technologies). Nuclei were counterstained with DAPI (Vector Laboratories). For tissue immunohistochemical staining, mice were sacrificed and intracardially perfused with 4% paraformaldehyde for 10 min. Mouse brain was removed and fixed overnight at 4°C, and dehydrated in a graded ethanol series and processed for paraffin embedding. For immunofluorescence, sections were incubated with the primary antibodies overnight at 4°C, and then fluorescent secondary antibodies with DAPI nuclear staining. The specificity of the staining was confirmed by omitting the primary antibodies. For DAB staining, the images of immunohistochemical results were obtained by a DMR-X microscope coupled with a DC500 digital camera (Leica) and the image analysis system Quantimet Q550 (Leica). For immunofluorescent double-labeled staining, the images were collected using a Leica SP5 confocal

microscope (Leica) and recorded sequentially using Leica Application Suite Software (Leica). All experiments were performed three or more times, and representative results are shown in the figures.

## In vivo model of brain ischemia and damage evaluation

This study was performed in strict accordance with the recommendations in the Guide for the Care and Use of Laboratory Animals of the National Institutes of Health. All of the animals were handled according to approved institutional animal care and use committee protocols of the Air Force Medical University. The protocol was approved by the Committee on the Ethics of Animal Experiments of Air Force Medical University (IR-2384-3915-3322). All surgeries were performed under sodium pentobarbital anesthesia, and every effort was made to minimize suffering. Then, 8- to 12-week-old mice (25–30 g) were obtained from the Laboratory Animal Center of the Air Force Medical University. MCAO was used to induce focal cerebral ischemia. Briefly, mice were anesthetized using 5% isoflurane in 30% $O_2$/70% $N_2O$ and placed in the supine position on a heating pad. Ischemia was produced by advancing the tip of a rounded nylon suture into the left internal carotid artery through the left external carotid artery. After 60 min of occlusion, the thread was withdrawn to allow reperfusion. Mice were sutured and placed in a 35°C nursing box to recover from anesthesia, and then returned to the cage. Sham group mice received midline neck incisions. The left common carotid artery was isolated, but not cut. In the MCAO + EPZ groups, mice were intranasally treated with 20 mg/kg EPZ015666 1 hr before surgery. The MCAO group mice were intranasally administered with an equal volume of the DMSO solution used to dissolve EPZ. After the animal was awakened, neurological damage was assessed as follows (*Kofler et al., 2006*): 0, no deficit; 1, forelimb weakness and torso turning to the ipsilateral side when held by tail; 2, circling to the affected side; 3, unable to bear weight on the affected side; and 4, no spontaneous locomotor activity or barrel rolling. If no deficit was observed 60 min after beginning of occlusion period, the animal was removed from further study.

After 24 hr reperfusion, the mice were sacrificed by rapid decapitation under deep anesthesia. Whole brain was rapidly removed for wet weight quantification, and then desiccated at 105°C for 48 hr until the weight was constant for dry weight quantification. The water content and tissue swelling were calculated as follows based on *Keep et al., 2012*:

$$\text{Water content ration} = \left(\text{Wet weight} - \text{Dry weight}\right)/\text{Dry weight}$$

$$\text{Initial wet weight}_{\text{ipsilateral}} = \text{Dry weight}_{\text{ipsilateral}} \times (\text{Water content}_{\text{contralateral}} + 1)$$

$$\%\,\text{Tissue swelling} = 100 \times (\text{Wet weight}_{\text{ipsilateral}} - \text{Initial wet weight}_{\text{ipsilateral}})/\text{Initial wet weight}_{\text{ipsilateral}}$$

Brain infarct area was evaluated using 2,3,5-triphenyltetrazolium chloride (TTC) staining. Brains were sectioned into 2-mm-thick coronal slices and stained in 2% TTC at 37°C for 15 min in the dark and then photographed. The infarct tissue areas were measured using Image-Pro Plus. To account for edema, the infarcted area was estimated by subtracting the uninfarcted region in the ipsilateral hemisphere from the contralateral hemisphere, and the infarct volume was expressed as a percentage of the contralateral hemisphere.

## ChIP and high-throughput sequencing

ChIP was performed as described previously (*Li et al., 2017*). Briefly, HT-22 cells were subjected to OGD treatment and crosslinked with 1% formaldehyde for 10 min at room temperature. Formaldehyde was quenched by 200 mM glycine and cells were rinsed twice with ice-cold PBS. Cells were transferred to 15 mL conical tubes and collected by centrifugation. Cells were lysed with lysis buffer A (50 mM HEPES-KOH [pH 7.5], 140 mM NaCl, 1 mM EDTA, 0.5% NP-40, 0.25% Triton X-100, 10% glycerol, and protease inhibitor cocktail [Roche]), incubated at 4°C for 10 min, and collected by spinning at 1300 × g for 5 min at 4°C. Cells were then resuspended in lysis buffer B (10 mM Tris–Cl [pH 8], 200 mM NaCl, 1 mM EDTA, 0.5 mM EGTA, and protease inhibitor cocktail), incubated at room temperature for 10 min. Nuclei were pelleted by spinning at 1300 × g for 5 min at 4°C. The pellet was suspended with lysis buffer B (10 mM Tris–Cl [pH 8], 100 mM NaCl, 1 mM EDTA, 0.5 mM EGTA, 0.1% Na-deoxycholate, 0.5% N-lauroylsarcosine, and protease inhibitor cocktail) and incubated for 15 min on ice. Chromatin shearing was conducted with cells on ice using a microtip attached to Misonix 3000 sonicator. Sonicate 8–12 cycles of 30 s ON and 90 s OFF around 30-watt power output. A final concentration of 1% Triton X-100 was added and gently mixed by pipetting. The chromatin solution was

clarified by spinning at 20,000 × *g* at 4°C for 30 min. Chromatin immunoprecipitation was performed with 50 uL Dynabeads protein G (Life Technology) conjugated preliminary antibodies antibody overnight at 4°C. The immunoprecipitated material was washed five times with wash buffer (10 mM Tris–Cl [pH 8], 1 mM EDTA, 0.5% NP40, 0.5 M LiCl, 0.5% Na-deoxycholate) and once with TE buffer (pH 8.0), then, eluted by heating for 30 min at 65°C with elution buffer (50 mM Tris–Cl [pH 7.5], 10 mM EDTA, 1% sodium dodecyl sulfate). To reverse the crosslinks, samples were incubated at 65°C overnight, then the eluted was digested with a final concentration of 0.5 µg/mL RNasesA at 37°C, followed by a final concentration of 0.5 µg/mL Proteinase at 55°C for 2 hr. The immunoprecipitated DNA were then purified using the DNA clean and concentrator 5 column (Zymo Research). The ChIP DNA was used for qPCRs using the indicated primers (*Supplementary file 3*), and data were plotted as the percentage of input. All experiments were performed three or more times, and representative results are shown in the figures. For ChIP-seq, 1 ng precipitated DNA or input was used to generate sequencing libraries using the Nextera XT DNA sample preparation Kit (Illumina). The resulting libraries were sequenced on Next-Seq (Illumina). Two biological replicates were performed, and combined reads were used for further analysis.

### Bioinformatics analysis

For ChIP-seq, reads were filtered if they had a mean Phred quality score of less than 20. They were aligned to the mm10 assembly using Bowtie v1.2 with parameters -v 2 -m 1 `--best --strata`. Reads that aligned to the same genomic coordinates were considered duplicates and removed using the MarkDuplicates tool in the Picard Tools suite v1.86. Fragment length estimates were obtained using Homer v4.3. Coverage tracks were generated using the genomecov tool in the BEDtools suite after extending the aligned reads to the estimated fragment lengths. Coverage tracks were normalized to coverage per 10 million mapped reads. Peaks were called using SICER v1.1 and the following parameters: species = mm10, redundancy threshold = 100, window size = 200, effective genome fraction = 0.77, gap size = 600, FDR = 0.000001. Peaks were labeled as 'TSS' if they overlapped with the TSS of a gene in the GENCODE vM12 annotation, and they were labeled as 'gene body' if they did not overlap with a TSS but did overlap with any other part of a gene body. They were labeled as 'intergenic' if they did not overlap with either of the previous two.

For RNA-seq, reads were filtered if they had a mean Phred quality score of less than 20. They were aligned to the mm10 assembly using STAR v2.6.0c. Gene counts were obtained using the featureCounts tool in the Subread package v1.5.1 with the GENCODE vM12 annotation. Differentially expressed genes (DEGs) were identified using DESeq2 with the following model: ~KnockDown + Treatment + KnockDown:Treatment. DEGs were required to have an FDR of less than 0.05 and a fold change of 1.5 or greater. 'Down Reversed Genes' were those that had a significant negative fold change in the OGD vs. control contrast and also a significant positive fold change in the interaction term contrast. The enrichment plot was generated using GSEA v.3.0.

### Statistical analysis

All data were presented as means ± SEM. Statistical analyses were performed using Student's *t*-test, one-way ANOVA, or two-way repeated-measures ANOVA followed by Bonferroni's multiple-comparison tests where appropriate. A value of $p < 0.05$ was considered significant.

### Acknowledgements

This work was supported by internal foundation (21QNPY084).

## Additional information

### Funding

| Funder | Grant reference number | Author |
| --- | --- | --- |
| Air force medical university | 21QNPY084 | Jiaji Lin |

| Funder | Grant reference number | Author |
|--------|------------------------|--------|

The funders had no role in study design, data collection and interpretation, or the decision to submit the work for publication.

## Author contributions

Haoyang Wu, Software, Funding acquisition, Investigation, Visualization; Peiyuan Lv, Data curation, Formal analysis, Supervision, Visualization, Methodology; Jinyu Wang, Resources, Data curation, Software, Funding acquisition, Methodology; Brian Bennett, Software, Validation, Visualization, Methodology, Project administration; Jiajia Wang, Data curation, Software, Formal analysis, Supervision, Funding acquisition; Pishun Li, Software, Formal analysis, Supervision, Funding acquisition, Investigation, Methodology; Yi Peng, Validation, Investigation, Methodology, Project administration; Guang Hu, Resources, Validation, Methodology, Writing – review and editing; Jiaji Lin, Conceptualization, Resources, Data curation, Software, Supervision, Validation, Visualization, Writing - original draft, Writing – review and editing

## Author ORCIDs

Jiaji Lin (ID) http://orcid.org/0000-0002-3653-2389

## Ethics

This study was performed in strict accordance with the recommendations in the Guide for the Care and Use of Laboratory Animals of the National Institutes of Health. All of the animals were handled according to approved institutional animal care and use committee protocols of Air Force Medical University. The protocol was approved by the Committee on the Ethics of Animal Experiments of Air Force Medical University. All surgery was performed under sodium pentobarbital anesthesia, and every effort was made to minimize suffering.

Reviewer #1 (Public Review): https://doi.org/10.7554/eLife.89754.2.sa1
Reviewer #2 (Public Review): https://doi.org/10.7554/eLife.89754.2.sa2
Author Response https://doi.org/10.7554/eLife.89754.2.sa3

# Additional files

## Supplementary files

• Supplementary file 1. A total of 125 genes involved in epigenetic and chromatin regulation were selected and used to generate an shRNA library in the pLKO.1 vector.
• Supplementary file 2. The PRMT5ChIP-seq were carried out in HT-22 cells treated with OGD, when intercepted with their RNA-seq data.
• Supplementary file 3. Primers and oligos used in this study.
• Supplementary file 4. Primary and secondary antibodies used in this study.
• MDAR checklist

## Data availability

Sequencing data are stored at https://www.ncbi.nlm.nih.gov/geo/query/acc.cgi?acc=GSE248393, including ChIP-Seq data (https://www.ncbi.nlm.nih.gov/geo/query/acc.cgi?acc=GSE248390) and RNA-Seq data (https://www.ncbi.nlm.nih.gov/geo/query/acc.cgi?acc=GSE248392).

The following datasets were generated:

| Author(s) | Year | Dataset title | Dataset URL | Database and Identifier |
|-----------|------|---------------|-------------|--------------------------|
| Lin J, Hu G, Bennett B | 2024 | Genetic Screen Identified Prmt5 as a Neuroprotection Target against Cerebral Ischemia | https://www.ncbi.nlm.nih.gov/geo/query/acc.cgi?acc=GSE248393 | NCBI Gene Expression Omnibus, GSE248393 |

*Continued*

| Author(s) | Year | Dataset title | Dataset URL | Database and Identifier |
|---|---|---|---|---|
| Lin J, Hu G, Bennett B | 2024 | Genetic Screen Identified Prmt5 as a Neuroprotection Target against Cerebral Ischemia [ChIP-seq] | https://www.ncbi.nlm.nih.gov/geo/query/acc.cgi?acc=GSE248390 | NCBI Gene Expression Omnibus, GSE248390 |
| Lin J, Hu G, Bennett B | 2024 | Genetic Screen Identified Prmt5 as a Neuroprotection Target against Cerebral Ischemia [RNA-seq] | https://www.ncbi.nlm.nih.gov/geo/query/acc.cgi?acc=GSE248392 | NCBI Gene Expression Omnibus, GSE248392 |

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
