## [Editor Report · eLife assessment]

The authors performed a **useful** RNAi screen to identify epigenetic regulators involved in oxygen-glucose deprivation (OGD)-induced neuronal injury. PRMT5 was identified as a negative regulator of neuronal cell survival after OGD. **Solid** in vitro and in vivo data suggest that PRMT5 could be a novel therapeutic target for the treatment of ischemic stroke.

---

## [Referee Report · Reviewer #1 (Public Review)]

The authors performed an RNAi screen to identify epigenetic regulators involved in oxygen-glucose deprivation (OGD)-induced neuronal injury using immortalized mouse hippocampal neuronal cell line HT-22. They identified PRMT5 as a novel negative regulator of neuronal cell survival after OGD. Both in vitro and in vivo experiments were then performed to evaluate the roles of PRMT5 in OGD and ischemic stroke-induced injury. The authors found that genetic and pharmacological inhibition of PRMT5 protected against neuronal cell death in both in vitro and in vivo models. Furthermore, they found that in response to OGD and ischemia, PRMT5 was translocated from the cytosol to the nucleus, where PRMT5 bound to the chromatin and promoter regions of targeted genes to repress the expression of downstream genes. Further, they showed that silencing PRMT5 significantly altered the OGD-induced changes for a large-scale of genes. In a mouse model of middle cerebral artery occlusion (MCAO), PRMT5 inhibitor EPZ015666 protected against neuronal death in vivo. This study reveals a potential therapeutic target for the treatment of ischemic stroke. Overall, the authors have done elegant work showing the role of PRMT5 in neuronal cell survival. However, the essential mechanisms underlying PRMT5 nuclear translocation have not been investigated, and the in vivo animal studies should be further strengthened.

---

## [Referee Report · Reviewer #2 (Public Review)]

Haoyang Wu et al. have shown that the symmetric arginine methyltransferase PRMT5 binds to the promoter region of several essential genes and represses their expression, leading to neuronal cell death. Knocking down PRMT5 in HT-22 cells by shRNA leads to pertinent improvement in cell survival after oxygen-glucose deprivation (OGD) conditions. In another set of experiments, inhibition of the catalytic activity of PRMT5 by a specific inhibitor, EPZ015666, in a middle cerebral artery occlusion (MCAO) mice model also showed protective effects against neuronal cell death. In this manuscript, the authors have established the negative role of PRMT5 in cerebral ischemia both in vitro and in vivo.

However, my primary concern is the novelty of the manuscript. It has already been reported that inhibition of PRMT5 attenuates cerebral ischemia/reperfusion condition (Inhibition of PRMT5 attenuates cerebral ischemia/reperfusion-induced inflammation and pyroptosis through suppression of NF-κB/NLRP3 axis. Xiang Wu et al. Neuroscience Letters, Volume 776, 2022, 136576, ISSN 0304-3940, https://doi.org/10.1016/j.neulet.2022.136576). Even these authors have also shown that treatment of PRMT5 specific catalytic inhibitor, LLY-283, could rescue ischemia-induced over-expression of inflammation-related factors.

However, it would be better to verify the specificity of the inhibitor, EPZ015666, using other methyltransferases to be sure that the rescue is indeed mediated by PRMT5 catalytic inhibition.

---

## [Author Response]

**Reviewer #1 (Public Review):**
The authors performed an RNAi screen to identify epigenetic regulators involved in oxygen-glucose deprivation (OGD)-induced neuronal injury using immortalized mouse hippocampal neuronal cell line HT-22. They identified PRMT5 as a novel negative regulator of neuronal cell survival after OGD. Both in vitro and in vivo experiments were then performed to evaluate the roles of PRMT5 in OGD and ischemic stroke-induced injury. The authors found that genetic and pharmacological inhibition of PRMT5 protected against neuronal cell death in both in vitro and in vivo models. Furthermore, they found that in response to OGD and ischemia, PRMT5 was translocated from the cytosol to the nucleus, where PRMT5 bound to the chromatin and promoter regions of targeted genes to repress the expression of downstream genes. Further, they showed that silencing PRMT5 significantly altered the OGD-induced changes for a large-scale of genes. In a mouse model of middle cerebral artery occlusion (MCAO), PRMT5 inhibitor EPZ015666 protected against neuronal death in vivo. This study reveals a potential therapeutic target for the treatment of ischemic stroke. Overall, the authors have done elegant work showing the role of PRMT5 in neuronal cell survival. However, the essential mechanisms underlying PRMT5 nuclear translocation have not been investigated, and the in vivo animal studies should be further strengthened.

Thank you very much for your comments and suggestions. While stroke stands as the second leading cause of death globally, and the burden of post-onset disability is substantial, particularly surging at a faster rate in low- and middle-income countries compared to high-income countries. The exploration of new drugs for stroke treatment holds profound societal implications. The concept of neuroprotective drug development is not novel; over the past half-century, considerable research and resources have been invested in this field. Yet, progress appears to be notably limited, and interest is currently waning.

Our research team is dedicated to devising rapid and cost-effective functional screening strategies grounded in the nervous system. Through this forward research approach, we aim to delve into potential neuroprotective targets across various neurological diseases. This endeavor not only bears significance for acute stroke but also holds potential application value for a spectrum of generalized nerve injuries.

Building on your insights, our upcoming studies will involve in vivo animal experiments, integrating the PRMT5 nuclear translocation mechanism. We anticipate that our continued research will benefit from further professional insights and guidance from your expertise.

**Reviewer #2 (Public Review):**
Haoyang Wu et al. have shown that the symmetric arginine methyltransferase PRMT5 binds to the promoter region of several essential genes and represses their expression, leading to neuronal cell death. Knocking down PRMT5 in HT-22 cells by shRNA leads to pertinent improvement in cell survival after oxygen-glucose deprivation (OGD) conditions. In another set of experiments, inhibition of the catalytic activity of PRMT5 by a specific inhibitor, EPZ015666, in a middle cerebral artery occlusion (MCAO) mice model also showed protective effects against neuronal cell death. In this manuscript, the authors have established the negative role of PRMT5 in cerebral ischemia both in vitro and in vivo.However, my primary concern is the novelty of the manuscript. It has already been reported that inhibition of PRMT5 attenuates cerebral ischemia/reperfusion condition (Inhibition of PRMT5 attenuates cerebral ischemia/reperfusion-induced inflammation and pyroptosis through suppression of NF-κB/NLRP3 axis. Xiang Wu et al. Neuroscience Letters, Volume 776, 2022, 136576, ISSN 0304-3940, https://doi.org/10.1016/j.neulet.2022.136576). Even these authors have also shown that treatment of PRMT5 specific catalytic inhibitor, LLY-283, could rescue ischemia-induced over-expression of inflammation-related factors.However, it would be better to verify the specificity of the inhibitor, EPZ015666, using other methyltransferases to be sure that the rescue is indeed mediated by PRMT5 catalytic inhibition.

Thank you sincerely for dedicating time from your busy schedule to review our papers. Your comments and suggestions hold immense value for us, contributing significantly to the enhancement of our work. We acknowledge with honesty that this research journey has been a prolonged and challenging experience.

The major functional study, as indicated by the CHIP-seq data record, was concluded between 2017 and 2019. Since then, our efforts and resources have been devoted to conducting in-depth mechanism and regulation research for PRMT5. Notably, PRMT5 is involved in 4-5 types of histone arginine methylation, and it plays a role in complex modification effects for proteins in the cytoplasm. Despite employing a variety of investigative methods, understanding and controlling these intricate mechanisms in experimental design have proven quite challenging. This not only places us at a disadvantage compared to some competitors but also hinders the creative potential of our lab team.

We firmly believe that there is ample room for further research on the role of PRMT5 in the nervous system. We aspire to collaborate with other research teams to explore this area collectively.